# Non-Ionic Osmotic Stress Induces the Biosynthesis of Nodulation Factors and Affects Other Symbiotic Traits in *Sinorhizobium fredii* HH103

**DOI:** 10.3390/biology12020148

**Published:** 2023-01-18

**Authors:** Francisco Fuentes-Romero, Isamar Moyano-Bravo, Paula Ayala-García, Miguel Ángel Rodríguez-Carvajal, Francisco Pérez-Montaño, Sebastián Acosta-Jurado, Francisco Javier Ollero, José-María Vinardell

**Affiliations:** 1Departamento de Microbiología, Facultad de Biología, Universidad de Sevilla, 41012 Seville, Spain; 2Departamento de Química Orgánica, Facultad de Química, Universidad de Sevilla, 41012 Seville, Spain; 3Centro Andaluz de Biología del Desarrollo, CSIC/Junta de Andalucía, Departamento de Biología Molecular e Ingeniería Bioquímica, Universidad Pablo de Olavide, 41013 Seville, Spain

**Keywords:** *Sinorhizobium fredii*, osmotic stress, rhizobia-legume symbiosis, transcriptomics, Nod factors, indole acetic acid, bacterial motility, acyl-homoserine lactones

## Abstract

**Simple Summary:**

Rhizobia are soil proteobacteria able to establish nitrogen-fixing symbiosis with host legumes. This symbiotic interaction, which is highly important from ecological and agronomical points of view since it allows growth of legumes in soils poor in nitrogen, requires a complex interchange of molecular signals between both symbionts. The production of rhizobial molecular signals is elicited by flavonoids, which are phenolic compounds exuded by legume roots. Recent work has shown that osmotic stress can also promote the formation of symbiotic signals in some rhizobia. In this work, we show that this is also the case for *Sinorhizobium fredii* HH103, a rhizobial strain able to establish symbiosis with hundreds of legumes, including the very important crop, soybean. Non-ionic osmotic stress, which can be encountered by the bacterium in the rhizosphere or inside the legume host, affected the expression of hundreds of bacterial genes and, consequently, influenced diverse bacterial traits, including the production of symbiotic signals and certain characteristics that may be relevant for successful interaction with the host: motility, production of the phytohormone indole acetic acid, and production of molecules involved in bacterial cell-to-cell communication. Thus, our work provides new evidence of how stress can promote rhizobia-legume symbiosis.

**Abstract:**

(1) Background: Some rhizobia, such as *Rhizobium tropici* CIAT 899, activate nodulation genes when grown under osmotic stress. This work aims to determine whether this phenomenon also takes place in *Sinorhizobium fredii* HH103. (2) Methods: HH103 was grown with and without 400 mM mannitol. β-galactosidase assays, nodulation factor extraction, purification and identification by mass spectrometry, transcriptomics by RNA sequencing, motility assays, analysis of acyl-homoserine lactones, and indole acetic acid quantification were performed. (3) Results: Non-ionic osmotic stress induced the production of nodulation factors. Forty-two different factors were detected, compared to 14 found in the absence of mannitol. Transcriptomics indicated that hundreds of genes were either activated or repressed upon non-ionic osmotic stress. The presence of 400 mM mannitol induced the production of indole acetic acid and acyl homoserine lactones, abolished swimming, and promoted surface motility. (4) Conclusions: In this work, we show that non-ionic stress in *S. fredii* HH103, caused by growth in the presence of 400 mM mannitol, provokes notable changes not only in gene expression but also in various bacterial traits, including the production of nodulation factors and other symbiotic signals.

## 1. Introduction

Rhizobia-legume symbiosis involves interactions mediated by molecular dialogue between prokaryotic and eukaryotic cells. In this signal exchange, flavonoids secreted by the roots of leguminous plants are key molecules. These phenolic compounds are believed to interact with the bacterial protein NodD, which belongs to the LysR family of bacterial transcriptional regulators, triggering the transcription of so-called nodulation (*nod*) genes by interacting with conserved promoter sequences, called *nod* boxes [1,2,3]. Some *nod* genes are responsible for the biosynthesis and secretion of another fundamental signal for the establishment of the symbiosis, nodulation factors (Nod factors, NFs). NFs, also called lipochitooligosaccharides or LCOs, are N-acetyl-glucosamine (GlcNAc) oligomers (3–6 residues) exhibiting different chemical decorations [1,4]. NFs are recognised by plant receptors of the LYSM type, initiating the whole reaction cascade that culminates in the formation of nitrogen-fixing nodules in the legume roots [2,3].

Molecular recognition between NFs and plant receptors is a key factor not only for the establishment of symbiosis but also for the specificity of the interaction between rhizobia and legumes. However, other bacterial molecular signals also play important roles in the symbiotic process [1,5,6]. Different surface polysaccharides function either as signal molecules required for the progression of symbiosis or as protective molecules that face the plant defence system. The rhizobial surface polysaccharides showing more symbiotic relevance include exopolysaccharides (EPS), lipopolysaccharides (LPS), K-antigen capsular polysaccharides (KPS) and cyclic glucans (CG) [1,7]. In some rhizobia, such as *Sinorhizobium fredii* and different species of the *Bradyrhizobium* genus, flavonoids and NodD also induce the expression of TtsI, the transcription of which is driven by a *nod* box. TtsI is the transcriptional activator of genes (*rhc*, *nop*) related to the assembly of a symbiotic type III secretion system (T3SS), which is responsible for the delivery of effector proteins (nodulation outer proteins, Nops) into host cells [1]. These effector proteins, called T3Es, can have positive, negative, or neutral effects on symbiosis depending on the rhizobial-legume pair [8]. In addition, the expression of some other rhizobial genes may be driven by *nod* boxes and thus activated by Nod and flavonoids. This is the case for genes involved in the production of the phytohormone indole acetic acid (IAA) in both *Sinorhizobium fredii* HH103 and *Rhizobium tropici* CIAT 899 [9]. Finally, flavonoids also manipulate quorum-sensing systems, activating *N*-acyl homoserine lactone (AHL) synthesis genes in some rhizobia, such as *S. fredii* SMH12, *Rhizobium etli* ISP42, or *Rhizobium sullae* IS123 [10].

During their free-living stage in the soil, rhizobia are exposed to multiple physical stresses, such as high temperatures, acidity, or elevated osmolarity. Interestingly, in *R. tropici* CIAT 899, in addition to flavonoids, some other molecules or conditions can activate the biosynthesis of NFs. Thus, this strain, which nodulates the common bean, can produce NFs not only in the presence of flavonoids but also under osmotic stress, either ionic (300 mM NaCl) or non-ionic (400 mM mannitol) [9,11]. In *R. tropici* CIAT 899, in addition to *nod* genes, non-ionic osmotic stress also activates the expression of nitrogen fixation genes and represses genes involved in motility or polysaccharide production [12]. Different studies indicate that *R. tropici* CIAT 899 has two different circuits for *nod* gene activation and Nod factor biosynthesis. The first is the classical flavonoid-mediated activation involving NodD1, whereas the second relies on osmotic stress activation and requires both NodD2 and OnfD, an AraC-type transcriptional regulator [13,14,15,16].

*Sinorhizobium fredii* HH103 is a fast-growing rhizobial strain that nodulates dozens of legume genera, including *Glycine max* (soybean). This strain was isolated from a soil sample from the Honghu county of Central China and was first described in 1985 [17]. *S. fred*ii strain HH103 can nodulate American and Asiatic soybean cultivars and is the best studied fast-growing soybean-nodulating strain to date [18]. Legumes nodulated by *S. fredii* HH103 can form determinate nodules, such as *Glycine max* and *Vigna unguiculata*, or indeterminate nodules, such as *Glycyrrhiza uralensis* and *Sophora tomentosa*. The most abundant Nod factors produced by HH103 consist of a mix of tri-, tetra-, and pentasaccharides carrying a saturated or unsaturated C16 or C18 fatty acyl residue in the non-reducing end of the molecule and a fucose or methylfucose modification at the C6 position of the reducing GlcNAc residue [19]. *S. fredii* HH103 symbiotic genes are regulated in a complex manner. In the presence of effective flavonoids such as genistein, NodD1 acts as the master activator of approximately 100 genes, including those related to Nod factor production and T3SS functioning [9,20,21]. However, other regulatory proteins participate in the modulation of symbiotic gene expression, such as the global repressor protein NolR, LysR family regulators NodD2 and SyrM, and the zinc-finger transcriptional regulator MucR1, all of which have different positive, neutral, or negative effects on expression depending on the gene [6,22,23,24,25].

The influence of additional *S. fredii* HH103 symbiotic signals during the nodulation process has also been investigated. The production of surface polysaccharides in *S. fredii* HH103 has been extensively described. In fact, the chemical structures of EPS, LPS, KPS, and CG are known, many of the genes involved in their production have been characterised, and the effect of the absence of any of these polysaccharides in symbiosis with soybean (and some other host legumes) has been studied [1,26,27]. A similar landscape emerges from reports dealing with T3SS and QS in this rhizobium, since both systems have been deeply described and the role of both Nops and AHL molecules during the symbiotic process has been analysed in detail [1,28,29,30,31,32,33,34].

In this work, we studied how non-ionic stress caused by the presence of a high concentration of mannitol affected *S. fredii* HH103 gene expression and whether this growth condition affected Nod factors production and different physiological traits, such as motility and production of surface polysaccharides IAA or AHLs.

## 2. Materials and Methods

### 2.1. Bacterial Strains and Plasmids

*S. fredii* HH103 Rif^R^ and its *nodD1*::Ω (= SVQ318) [35] and *nodD2*::Ω (= SVQ515) [34] were grown at 28 °C on tryptone yeast (TY) medium [36]. For the β-galactosidase assay, AHL production, and RNA-seq analysis, yeast extract mannitol (YM) medium [37] was used. B^−^ minimal medium [38] was used for NF-related experiments. Minimal medium (MM) [39] was used for the surface motility and IAA assays. All of these media, except for TY, contained 10 g L^−1^ mannitol (55 mM). Swimming motility was studied in Bromfield medium (BM) (0.04% tryptone, 0.01% yeast extract, and 0.01% CaCl_2_ 2H_2_O) [40]. When required, media were supplemented with mannitol or the flavonoid genistein at a final concentration of 400 mM or 3.7 µM, respectively.

*Agrobacterium tumefaciens* NT1 (pZLR4), carrying a *traG*::*lacZ* fusion and *traR* for producing blue colour in the presence of X-Gal (5-bromo-4-chloro-3-indolyl-β-D-galactopyranoside) and appropriate AHLs, was grown at 28 °C in YM supplemented with 30 µg mL^−1^ gentamicin [41].

The growth curves of *S. fredii* HH103 in different media were obtained using a Sinergy HT microplate reader (Agilent, Santa Clara, CA, USA) for 72 h at 28 °C with continuous orbital shaking. Measurements were undertaken every 4 h.

### 2.2. β-Galactosidase Assay

The β-galactosidase assay was carried out as previously described [20,42]. This assay was carried out with *S. fredii* HH103 (pMP240) grown in YM medium with or without 400 mM mannitol or 3.7 µM genistein. Plasmid pMP240 belongs to the IncP incompatibility group and harbours a Tc^R^ gene and transcriptional fusion occurs between the *R. leguminosarum* bv. *viciae nodA* promoter and a *lacZ* gene lacking its own promoter [43]. Bacterial cultures at OD_660_ = 0.8–1.0 were diluted (approximately 100-fold) before culturing in YM and YM supplemented with 400 mM mannitol to ensure that the cultures had an OD_660_ in the range of 0.15–0.30 when β-galactosidase activity was measured (approximately 16 h later).

### 2.3. Nod Factor Purification, Biological Activity, and Structural Determination by UHPLC-HRMS/MS

Nod factors (NFs) were purified from 1 L of culture for each sample following the previously described procedure [11]. The purified NFs were resuspended in 50 mL of acetonitrile 20% and 1 μL per mL of plant nutrient solution was added for the biological activity assays. *G. max* cv. Williams seeds were surface-sterilised and mounted in test tubes on a curled wire with the roots in 30 mL of Fåhraeus medium [44]. Roots were protected from light and plants were grown for 10 days. The growth chamber conditions were 16 h at 26 °C in light and 8 h at 18 °C in the dark, with 70% humidity. To determine the presence of nodule primordia, roots were cleared with sodium hypochlorite and stained with methylene blue using the previously described method [45]. Each experiment was repeated three times with six plants for each treatment.

To determine the chemical structure of the NFs, samples were analysed using an Ultra-High Performance Liquid Chromatography (UHPLC) system consisting of a quaternary UHPLC Dionex Ultimate 3000 SD (Thermo Fisher Scientific, Waltham, MA, USA) connected to a quadrupole-orbitrap Qexactive hybrid high-resolution mass spectrometer (HRMS) (Thermo Fisher Scientific, Waltham, MA, USA) with a HESI ionisation probe, as previously described [12]. The lists of NFs produced in each condition are provided in Appendix A.

### 2.4. RNA Extraction and Sequencing

*S. fredii* HH103 was grown in 50 mL Falcon tubes containing 7 mL of YM medium to ensure good aeration and supplemented with 400 mM mannitol when necessary. Bacteria were incubated with an orbital shaker (180 rpm) for 72 h at 28 °C, until reaching an OD_600_ of 0.8–1.0. Total RNA was isolated using a High Pure RNA Isolation Kit (Roche, Basel, Switzerland) according to the manufacturer’s protocol. Verification of the quantity and quality of the total RNA samples was carried out using a Nanodrop 1000 spectrophotometer (Thermo Scientific, Waltham, MA, USA) and a Qubit 2.0 Fluorometer (Invitrogen, Waltham, MA, USA). Two independent total RNA extractions were obtained for each condition with or without osmotic stress.

Ribosomal RNA was depleted using a MICROB Express Bacterial mRNA Purification kit (Ambion, Waltham, MA, USA) following the manufacturer’s protocol. The integrity and quality of the ribosomal RNA-depleted RNA were checked using an Agilent Bioanalyzer 2100 (Agilent Technologies, Santa Clara, CA, USA). RNA sequencing was carried out by Sistemas Genómicos ((Valencia, Spain) using the Illumina Next Generation Sequencing (NGS) platform with 100 bp paired-end reads using the Illumina HiSeq 2000 sequencing instrument (Illumina, San Diego, CA, USA). Ribosomal RNA-depleted samples were used to generate whole transcriptome libraries following the manufacturer’s recommendations for sequencing on this NGS platform. Amplified cDNA quality was analysed using the Bioanalyzer 2100 DNA 1000 kit (Agilent Technologies, Santa Clara, CA, USA) and quantified using a Qubit 2.0 Fluorometer (Invitrogen, USA). The RNA-Seq data discussed in this publication were deposited in the Sequence Read Archive of NCBI (BioProject database) under BioProject IDs PRJNA313151 [46] and PRJNA911807.

A total of four RNA-seq libraries corresponding to *S. fredii* HH103 under control and 400 mM mannitol conditions were generated (two independent biological experiments for each condition). Quality control of each run, sample normalisation, and statistical procedures were performed as previously described [9,46]. Differentially expressed genes for each strain and condition were obtained by comparing against gene expression levels of the wild-type strain grown under control conditions (Appendix A). The dataset was validated by *q*RT-PCR as previously described [9,46]. Briefly, the (DNA free) RNA extracted from 24-h old cultures was reverse transcribed to cDNA using the QuantiTec Reverse Transcription Kit (Qiagen, Hilden, Germany). Quantitative PCR was performed using a LightCycler 480 (Roche) with the following conditions: 95 °C, 10 min; 95 °C, 30 s; 50 °C, 30 s; 72 °C, 20 s; for 40 cycles, followed by melting from 60 °C to 95 °C to verify the specificity of the reaction. The 16S RNA gene was used as an internal control to normalize gene expression. The fold-change in gene expression was obtained using the ΔΔCt method [47]. Selected genes and primers are listed in Table 1.

### 2.5. RNA-seq Data Analysis

First, quality control of the raw data was carried out by FASTQ. Next, the initial whole transcriptome paired-end reads obtained from sequencing were mapped against the latest version of the NCBI HH103 genome: NC_016812 (Chromosome), NC_16813 (plasmid a1), NC_16814 (plasmid c), NC_16815 (plasmid e), NC_16836 (plasmid b), NC_16846 (plasmid d, part 1), NC_16847 (plasmid d, part 2), NC_16848 (plasmid d, part 3), NC_16849 (plasmid d, part4), and NC_16850 (plasmid d, part 5), using the mapping algorithm Bowtie2 v2.3.1 [48]. Low-quality reads were eliminated using Samtools [49,50] and Picard Tools (http://broadinstitute.github.io/picard/), leaving only high-quality reads. The genetic quantification was calculated using the htseq count 0.6.1p1 method [51]. Gene differential expression and quantification were obtained using DESeq2 [52]. Concordance between samples under the same conditions was determined by correlation and Euclidean distance using the statistic software R. To determine differential expression among groups of samples, Phyton and R software and the DESeq2 algorithm were used. Differentially expressed genes (DEGs) in this study were established as those genes with a fold-change lower or higher than −3.0 or 3.0, respectively, with a *p* value adjusted to 0.05.

### 2.6. Quantification of Indole Acetic Acid (IAA) Production

Quantification of IAA from *S. fredii* HH103 was carried out using the Salkowski colourimetric assay, as described by del Cerro et al. [13]. Briefly, 5 mL of TY medium with tryptophan (0.4 g L^−1^) (supplemented when required with 400 mM mannitol) was inoculated with *S. fredii* HH103 and incubated for 96 h at 28 °C with an orbital shaker (180 rpm). Of these cultures, 1 mL of each sample was centrifuged to remove cells. The cell-free culture supernatant was assayed for IAA production. Every experiment was performed three times with eight replicates each time.

### 2.7. Motility Assay

Swimming was examined on plates prepared with BM containing 0.3% Bacto agar and surface motility was analysed on semisolid MM plates containing 0.4% agarose. When necessary, plates were supplemented with mannitol, ethanol, or the inducer flavonoid genistein, at final concentrations of 400 mM, 0.1% *v*/*v*, and 3.7 µM, respectively. Preparation of plates, inoculation with *S. fredii* HH103, and calculation of the migration zone were carried out as previously described [53].

### 2.8. PAGE Analysis of LPS and KPS

LPS extraction from bacterial cultures grown in solid TY medium in the presence or absence of 400 mM mannitol, separation on SDS-PAGE gels, and silver staining were performed as previously described [54]. For visualisation by PAGE, KPS was extracted from bacterial cultures grown in solid TY medium in the presence or absence of 400 mM mannitol, according to Hidalgo et al. [54]. Samples were analysed by PAGE as described by Parada et al. [55], but absolute ethanol was added to the running buffer and running and stacking gels at a final concentration of 10% (*v*/*v*) in all cases. The acrylamide concentration of the running gel was 18% (*w*/*v*) and the acrylamide:N,N-methylenebisacrylamide ratio was 29:1. Gels were fixed using Alcian Blue (0.5% in 2% acetic acid) and silver-stained. At least two independent cultures of each strain were used for the extraction of LPS and KPS.

### 2.9. AHL Well Diffusion Assay

*A. tumefaciens* NT1 (pZRL4) was used for the detection of AHLs (acyl chains ranging from C4 to C18) from supernatants of the parental strain grown to the late stationary phase in a well diffusion assay in Petri dishes, as described by McClean et al. and Cha et al. [56,57].

### 2.10. HPLC and Mass Spectrometry Analysis of AHL

Rhizobia were grown in 500 mL of YM medium for six days at 28 °C with shaking. Cultures were centrifuged and supernatants were extracted with dichloromethane and the organic layer was evaporated. The extracts were dissolved in 1 mL of methanol:water (1:1 *v*/*v*) containing 0.1% (*v*/*v*) formic acid, microfiltered (0.2 µm), and 20 mL was injected into an HPLC system equipped with a Tracer Hypersyl ODS column: 250 × 4.6 mm, 5 µm particle size (Teknokroma, Sant Cugat del Vallès, Spain). Methods using high pressure liquid chromatography and tandem mass spectrometry (HPLC-MS/MS) were carryout out as previously described [33]. Mass spectrometric conditions were optimised by infusing solutions of standards dissolved in methanol (100 mg mL^−1^) at a flow between 10–100 mL min^−1^: C4-HSL, C6-HSL, 3-oxo-C6-HSL, C8-HSL, 3-oxo-C8-HSL, C10-HSL, 3-oxo-C10-HSL, C12-HSL, 3-oxo-C12-HSL, 3-OH-C12-HSL, C14-HSL, 3-oxo-C14-HSL, and 3-OH-C14-HSL. The probe capillary voltage was optimised at 5500 V. The desolvation temperature was set to 50 °C. The pressures of curtain, nebulising, and turbo spray gases were set to 35, 20, and 0 (arbitrary units), respectively. Nitrogen was used for CID (collision-induced dissociation). Ions were scanned from *m*/*z* 150 to 500 at a scan rate of 4000 Th s^−1^.

## 3. Results

### 3.1. Expression of S. fredii HH103 Nod Genes Is Induced by the Presence of 400 mM Mannitol

In previous works, we showed that non-ionic osmotic stress, 400 mM mannitol, induces *nod* gene expression and Nod factor production in *R. tropici* CIAT 899 [12]. In the present work, we wanted to investigate whether this also occurs in another rhizobial strain, *S. fredii* HH103. For this purpose, we analysed *lacZ* expression driven by the promoter of the *nodA* gene of *Rhizobium leguminosarum* bv. v*iciae* in *S. fredii* HH103 harbouring the pMP240 plasmid, when grown in the presence of different mannitol concentrations: from 55 mM (the concentration of mannitol present in YM medium) to 1 M. The expression of p*nodA*::*lacZ*Δp (measured as β-galactosidase activity) was significantly higher than that found in YM medium in the range of 100 to 800 mM mannitol, reaching a maximum of 2-fold at 300–500 mM mannitol (Appendix A). These results indicated that a high concentration of mannitol positively influenced *nod* gene expression, not only in *R. tropici* CIAT 899 [12] but also in *S. fredii* HH103. However, this effect was more limited than the presence of *nod* gene-inducing flavonoids such as genistein, which caused a 12-fold increase in p*nodA*::*lacZ*Δp expression (Figure 1).

Since *S. fredii* HH103 carries two different copies of *nodD* (*nodD1* and *nodD2*), we investigated if either gene mediated the observed increase in *nod* gene expression in the presence of 400 mM mannitol. For this purpose, we studied the expression of p*nodA*::*lacZ*Δp in the presence of 400 mM mannitol or 3.7 µM genistein in the wild-type and in *nodD1* and *nodD2 S. fredii* HH103 mutant backgrounds (Figure 1). As expected, genistein enhanced β-galactosidase activity in *S. fredii* HH103 *nodD2* but not in *S. fredii* HH103 *nodD1* since NodD1 is responsible for the flavonoid-induced expression of *nod* genes in this bacterium [20]. Interestingly, the presence of 400 mM mannitol did not enhance p*nodA*::*lacZ*Δp expression in either the *nodD1* or the *nodD2 S. fredii* HH103 mutants. This result suggested that both NodD1 and NodD2 may play a role in the induction of *nod* gene expression caused by the presence of non-ionic osmotic stress.

### 3.2. S. fredii HH103 Produces Biologically Active Nod Factors in the Presence of 400 mM Mannitol

As mentioned above, the enhanced expression of *R. tropici* CIAT 899 *nod* genes provoked by the presence of 400 mM mannitol resulted in an increase in the production of NFs [12]. In addition, these NFs were shown to be biologically active since they promoted the formation of nodule primordia in the roots of the host legume *Phaseolus vulgaris*.

The fact that the presence of 400 mM mannitol also increased *nod* gene expression in *S. fredii* HH103 prompted us to investigate whether Nod factor production was also affected by this growth condition. For this purpose, we carried out ultra-high performance liquid chromatography (UHPLC) followed by data-dependent high-resolution mass spectrometry (UHPLC–HRMS/MS) of *S. fredii* HH103 cultures grown in YM medium in the absence or presence of non-ionic osmotic stress (400 mM mannitol). Cultures of *S. fredii* HH103 grown in YM medium supplemented with 3.7 µM genistein were also included in this study as a control for NF production. As shown in Figure 2A, when *S. fredii* HH103 was grown in YM medium, 14 different NFs were detected, whereas the presence of genistein or high mannitol concentrations led to an increase in the number of Nod factors produced (124 and 42, respectively).

The complete list of Nod factors produced under each condition is provided in Appendix A. All the NFs produced in YM (14 NFs) were also found among those produced in the presence of either genistein or high mannitol concentrations (Figure 2B). In addition, 19 other NFs were also found in both inducing conditions, whereas 9 and 91 different NFs were exclusively found in the presence of either 400 mM mannitol or 3.7 µM genistein, respectively. Thus, a core of 33 different NFs was produced under both *nod* gene-inducing conditions. Among them, NFs having 3 to 5 GlcNAc residues and harbouring predominantly saturated or monounsaturated C16 or C18 acyl groups as well as fucose or methyl fucose decorations could be found (Appendix A).

The putative biological activity of the NFs produced by *S. fredii* HH103 upon growth in the presence of 400 mM mannitol was studied by adding NF extracts to germinated soybean seedlings. Thirteen days after inoculation, the roots were stained with methylene blue and examined to detect the possible presence of nodule primordia (Figure 3). In the case of non-induced *S. fredii* HH103 cultures (B^−^ medium, 55 mM mannitol), the purified NF extract slightly induced the formation of primordia (2.8 ± 0.9 primordia/root), a phenomenon that was previously described for *R. tropici* CIAT 899 [12,15] and might be because a small number of NFs could be detected in such extracts. However, when the purified extracts of *S. fredii* HH103 cultures grown in 400 mM mannitol were added to soybean seedlings, a significant increase (approximately 3-fold) in the formation of nodule primordia could be detected (7.6 ± 1.5 primordia/root). In fact, the number of nodule primordia induced under the latter condition was very similar to that obtained when NFs purified from genistein-induced *S. fredii* HH103 cultures were added (7.9 ± 1.0 primordia/root) to soybean roots. These results clearly indicated that the NFs produced upon growth in B^−^ supplemented with a final concentration of 400 mM mannitol were, like those produced upon induction with genistein, biologically active.

### 3.3. A High Concentration of Mannitol Affects the Expression of Hundreds of S. fredii HH103 Genes

In a previous work, it was determined that non-ionic stress caused by the presence of 400 mM mannitol altered the expression of a high number of genes and consequently affected different biological processes in *R. tropici* CIAT 899 [12]. To check whether this also takes place in *S. fredii* HH103, we performed RNA-seq analysis of *S. fredii* HH103 cultures grown in YM medium (55 mM mannitol) and YM supplemented with a final concentration of 400 mM mannitol. The number of differentially expressed genes (DEGs) with fold-changes ≥3 with mannitol compared to the control was 743, including 350 induced DEGs and 393 repressed DEGs (Appendix A). Since the genome of *S. fredii* HH103 is estimated to have 6960 genes [58], the presence of mannitol affected the expression of more than the 10% genome of this strain. Remarkably, the percentage of induced or repressed genes varied depending on the replicon (Table 2). For example, regarding the symbiotic plasmid (pSfHH103d), 77 genes were induced whereas only 6 were repressed. On the other hand, the number of repressed genes (292) was considerably higher than that of induced genes (119) in the chromosome. Regarding the largest plasmid (pSfHH103e), the number of induced genes was double the number of repressed genes (148 vs. 72). Among the different small plasmids present in *S. fredii* HH103, it is noteworthy that more than 20% of the genes present in plasmid pSfHH103a2 (8) were repressed whereas none of them were induced in the presence of 400 mM mannitol. Appendix A contain detailed lists of all the *S. fredii* HH103 DEGs sorted by their fold-change or genome location, respectively. In both cases, the annotation of each DEG is provided.

Transcription of 11 of the found DEGs was quantified by *q*RT-PCR to validate the dataset (Table 3). In all cases, except for *nodA*, the same type of change of expression (either induction or repression) upon growth with 400 mM mannitol was detected.

### 3.4. The Presence of a High Concentration of Mannitol Induces the Expression of 350 S. fredii HH103 Genes

Among the 350 induced genes in the presence of 400 mM mannitol, 91 genes (26.0%) corresponded to hypothetical proteins of unknown function. 38 genes were involved in ABC-type transport and another seven genes were involved in other types of transport of different metabolites. According to their predicted functions, these transport systems would be responsible for the uptake of sugars, such as glucose or fructose, dicarboxylic acids, nitrate, sulphate, glycine betaine, proline, or spermidine (Appendix A). Similarly, three genes coding for products with BON domains, which are found in proteins that protect cells from osmotic stress, were overexpressed in 400 mM mannitol. In addition, two genes coding for universal stress proteins, psfHH103d_567 (+4.2) and psfHH103d_586 (+4.1), one encoding a DnaJ-like protein (psfHH103d_507, +3.6), and two genes coding for GroEL chaperones, SFHH103_00877 (+4.1) and psfHH103d_587 (+3.1), were also induced.

Regarding genes related to symbiosis, eight *nif* or *fix* genes involved in the biological fixation of atmospheric nitrogen were induced, including *nifS*, *nifN*, *fixP*, *fixB* or *fixJ* (Appendix A). Other *nif* and *fix* important genes showed levels of induction lower than 3-fold, such as *nifHDK*, which codes for the different polypeptides constituting the nitrogenase enzyme, the levels of expression of which increased approximately 2-fold. In *S. fredii* HH103, two clusters of *nod* genes account for the production of NFs: the *nodABCIJnolO’noeI* (psfHH103d_126 to psfHH103d_132) operon and the *nolK-noeL-nodZ-noeK* and *noeJ* genes (psfHH103d_379 to psfHH103d_383) [35,58,59]. Surprisingly, none of these *nod* genes were scored as DEGs in the presence of 400 mM mannitol, although some of them displayed transcriptional activation higher than 2-fold (Appendix A). Instead, two copies of the *nodU* gene exhibited increased expression: SFHH103_05865 (+4.1), located on pSfHH103e, and the pseudogene 454 (+3.9), harboured by the symbiotic plasmid pSfHH103d. Different genes involved in the formation of the T3SS apparatus (*rhcJ*, *nopX*, and *rhcN*, among others), as well as genes coding for effectors secreted by this apparatus such as *nodD* and *gunA,* were also induced by the presence of 400 mM mannitol. In addition, different genes involved in the production of surface polysaccharides, such as *rkpI* and *rkpJ* (K-antigen capsular polysaccharide or KPS) [18,54] and *exoI*, *exoN,* or *exoF* genes (exopolysaccharide, EPS) [60], were induced under this condition. Finally, other important symbiotic genes, such as those responsible for the biosynthesis of indole acetic acid (IAA), psfHH103d_257 to psfHH103d_259, which are located on the pSym and are under the transcriptional control of a *nod* box [58], were strongly upregulated upon the high concentration of mannitol (Appendix A).

Other genes showing increased expression in the presence of 400 mM mannitol included: (i) *sinI,* responsible for the synthesis of long-chain acyl-homoserine lactones (AHLs) [33]; (ii) different genes involved in the synthesis of cytochromes and the enzymes NADH dehydrogenase and nitrate reductase; (iii) 12 genes coding for different types of transcriptional regulators, including members of the HTH, GntR, and XRE families and the negative regulator TraM.

### 3.5. The Presence of a High Concentration of Mannitol Represses the Expression of 393 S. fredii HH103 Genes

Among the 393 genes differentially repressed in the presence of 400 mM mannitol, 97 of them (24.7%) coded for hypothetical proteins of unknown function Similar to the induced genes, a high number of genes (18) involved in ABC-type transport processes (iron, zinc, cyclic glucans, methionine, phosphonate, sugars) were repressed. Other processes that could be affected by the repressed genes included cell division and chromosome or plasmid segregation (eight genes), bacterial cell wall biosynthesis (ten), or chemotaxis (eight).

Concerning genes related to stress responses, four different copies of the tandem genes coding for GroES/GroEL chaperones were strongly repressed (between −8.3 and −20.1-fold), in contrast to the two previously mentioned copies of GroEL, one located on the pSym and one on the chromosome, that were induced 3.1–4.0-fold. Two of the repressed copies were located on the chromosome (SFHH103_00476-SFHH103_00475, SFHH103_03405-SFHH103_03404), one on the plasmid c (SFHH103_04126-SFHH103_04125), and the other on plasmid a2 (psfHH103a2_17-psfHH103a2_16). Additionally, the chromosomal copy of *dnaJ* (SFHH103_00574) was strongly repressed (9.4-fold), in contrast to the copy harboured by the pSym, which was induced 3.6-fold. Five multidrug efflux pump genes, which are related to the exclusion of toxic substances from the bacterial cell, were also repressed.

Regarding genes involved in surface appendages, two genes required for flagellum biosynthesis, *fliP* and *flaC*, as well as several genes involved in type IV pili biosynthesis, including *pilQ* or *cpaA*, were repressed in the presence of 400 mM mannitol. Although four genes responsible for EPS biosynthesis (*exoN*, *exoO*, *exoI*, and *exoF3*) were induced in the presence of mannitol, *exoN*, *exoB,* a possible EPS transporter, and *exoR*, which codes for a negative regulator of EPS biosynthesis, were repressed. Another repressed gene was *grlR*, the product of which is predicted to act as a negative regulator of T3SS, a system that, as mentioned before, is activated in the presence of non-ionic osmotic stress. In contrast, four out of seven genes (*rhcV-2*, *rhcU-2*, *rhcT-2*, and *rhcC2-2*) involved in the production of a second T3SS apparatus (T3SS-2) were strongly repressed at the high concentration of mannitol (Appendix A).

In addition, a high number of genes coding for transcriptional regulators (at least 18) were repressed by the presence of 400 mM mannitol, including members of the LuxR, LysR, LacI, and GntR families, among others.

### 3.6. The Presence of a High Concentration of Mannitol Affects different Physiological Traits in S. fredii HH103 Genes

The transcriptomics analysis of the effect of non-ionic stress on *S. fredii* HH103 mentioned above showed differential expression of genes involved in different biological processes beyond NF production, T3SS formation, or nitrogen fixation, including those related to motility, surface polysaccharide biosynthesis, and production of IAA and AHLs (Appendix A). Therefore, we decided to investigate whether these processes were affected by the presence of 400 mM mannitol. Before studying these bacterial traits, we examined whether the growth of *S. fredii* HH103 might be altered by the presence of 400 mM mannitol in the different culture media that would be employed in those analyses (see Material and Methods). As shown in Appendix A, this non-ionic osmotic stress negatively affected *S. fredii* HH103 growth in TY and YM media and MM, but had a positive effect in BM.

#### 3.6.1. Motility

In a recent study, we demonstrated that *S. fredii* HH103 exhibited surface motility in the presence of genistein in a NodD1- and TtsI (master activator of the T3SS)-dependent manner. However, the presence of this *nod* gene-inducing flavonoid did not influence the swimming capacity of this strain [53]. Since the expression of some *S. fredii* HH103 flagellar genes was repressed by the presence of 400 mM mannitol, we decided to investigate whether this stress condition might affect *S. fredii* HH103 motility. For this purpose, we carried out swimming and surface motility assays in the presence or absence of 400 mM mannitol. As shown in Figure 4, the swimming ability of *S. fredii* HH103 was fully abolished in the presence of a high concentration of mannitol. In contrast, the presence of 400 mM mannitol promoted *S. fredii* HH103 surface motility, although to a lower extent than genistein (Figure 5).

As described earlier, genistein-induced surface motility of *S. fredii* HH103 is dependent on the presence of a functional T3SS [53]. To check whether this was also the case for surface motility induced by a high concentration of mannitol, this trait was studied in an HH103 *ttsI*::Ω, a mutant derivative of *S. fredii* HH103 that lacked the positive regulator of this system, TtsI [34]. The growth of this mutant slightly extended from the centre of the plate (Figure 5). However, this could have been a consequence of high EPS production due to the high amount of carbon source available rather than surface motility, which suggested that T3SS was also essential for surface motility when induced by non-ionic osmotic stress in *S. fredii* HH103.

Genistein-induced surface motility of *S. fredii* HH103 is mediated by a combination of flagellum-dependent and -independent mechanisms [53]. Thus, we also investigated whether the presence of flagella was required for *S. fredii* HH103 surface motility upon growth in the presence of 400 mM mannitol. The growth of HH103 Δ*flaCBAD*, a derivative of *S. fredii* HH103 that lacked flagella [26], slightly extended from the centre of the plate, which, again, could have been the result of increased EPS production rather than surface motility (Figure 5). This result suggested that the surface motility exhibited by HH103 upon non-ionic osmotic stress was mediated by a flagellum-dependent mechanism.

#### 3.6.2. Surface Polysaccharides Biosynthesis

The fact that the expression of various genes involved in surface polysaccharide biosynthesis was affected by non-ionic osmotic stress led us to investigate whether the production of some of those polysaccharides might be altered by the growth condition. Regarding EPS, as commented above, several *exo* genes were affected, some of which were induced, and others were repressed. When grown in YM medium, no obvious changes in the appearance of mucoid colonies could be scored (data not shown), but a clear increase in mucoidy was observed in the presence of 400 mM mannitol when grown in TY medium (Figure 6).

On the other hand, two genes involved in KPS production (*rkpIJ*) were induced and several genes putatively involved in LPS synthesis (*cpsA*, *SFHH103_02601*, *SFHH103_05300*, *SFHH103_05856*) were either induced or repressed when *S. fredii* HH103 was grown in the presence of 400 mM mannitol (Appendix A). However, the electrophoretic profiles of these two polysaccharides were apparently not affected by non-ionic osmotic stress (Appendix A).

#### 3.6.3. Production of IAA

In *S. fredii* HH103, the psfHH103d_257-psfHH103d_259 genes, which are under the transcriptional control of a *nod* box, are involved in IAA synthesis [46,58]. Since these genes were also induced by non-ionic osmotic stress, we analysed the production of IAA by *S. fredii* HH103 when grown in the presence or absence of a high concentration of mannitol. As expected, the presence of 400 mM mannitol significantly enhanced (2-fold) the production of IAA (45.9 ± 5.5 mg/L) compared to that under control conditions (22.9 ± 1.4 mg/L) (non-parametric test of Mann–Whitney, α = 1%).

#### 3.6.4. Production of AHLs

*S. fredii* HH103 presents two known genes (*traI* and *sinI*) coding for AHL synthases. TraI and SinI are responsible for the production of short-(C4 to C8) and long-(C10 to C16) acyl chain AHLs, respectively, but at very low, non-physiological concentrations [33,58]. According to the RNA-seq analysis, the *sinI* gene was highly upregulated in the presence of 400 mM mannitol (Appendix A). Thus, *S. fredii* HH103 supernatants from cultures grown with or without a high concentration of mannitol were inspected for AHL production in well diffusion assays using the biosensor *A. tumefaciens* NT1 (pZRL4). Interestingly, the presence of AHLs could only be detected under the non-ionic osmotic stress condition (Figure 7A).

In a previous work, qualitative mass spectrometry analysis using HPLC-MS/MS unequivocally identified C8-HSL and 3-oxo-C8-HSL, produced by TraI, and C12-HSL, C14-HSL, and 3-oxo-C14-HSL, synthesised by SinI, in supernatants of *S. fredii* HH103 cultures, despite long-chain AHLs being present as trace compounds [33]. Here, we performed quantitative mass spectrometry analysis, establishing correlations between signal areas of any particular AHL and its relative abundance in supernatants from cultures with or without high concentration of mannitol (Appendix A). Figure 7B shows the areas of AHL signals present in supernatants from *S. fredii* HH103 cultures grown in the presence or absence of 400 mM mannitol (only AHLs detected in four biological replicates with average area values ≥ 10^5^ are represented). Following this approach, quantitative variations for individual AHLs were estimated. We found that short-chain AHLs (3-oxo-C8-HSL and C8-HSL) were present at high concentrations in both control and mannitol supernatants. Conversely, the set of long-chain AHLs (3-oxo-C12-HSL, C12-HSL, 3-oxo-C14-HSL, C14-HSL, 3-oxo-C16-HSL, and C16-HSL) was only detected when *S. fredii* HH103 was grown with a high concentration of mannitol. Altogether, these results suggested that non-ionic osmotic stress conditions induced the production of long-chain AHLs by *sinI* transcriptional activation in *S. fredii* HH103.

## 4. Discussion

Rhizobia are soil bacteria that can shift between two different styles of life: free-living saprophytic bacteria in the soil and endosymbionts hosted inside specialised organs, called nodules, formed by the roots of legumes in response to the presence of compatible rhizobia [61]. Because of this, rhizobia have to survive in different environments and cope with different kind of stresses, including high temperatures, acidity, or basicity, as well as elevated osmolarity due to high concentrations of different solutes either in the soil or inside the host plant [62].

Non-ionic osmotic stress created by the presence of 400 mM mannitol had a substantial impact on both the bacterial growth and gene expression of *S. fredii* HH103. On the one hand, as expected, this condition negatively affected *S. fredii* HH103 growth in TY and YM media and MM, but, interestingly, it had a positive effect in BM, most likely because of the low osmolarity of this medium (it is basically a 10-fold diluted version of TY). On the other hand, growth upon 400 mM mannitol affected the expression of more than 10% of the annotated genes of *S. fredii* HH103, and the numbers of induced and repressed genes were similar (350 and 393 respectively). Remarkably, the percentages of induced and repressed genes varied greatly depending on the replicon (Table 2). In the case of the symbiotic plasmid, these percentages were 11.5% and 0.9% respectively, in contrast to the 0.0% and 21.1% found for plasmid pSfHH103a2. In the case of pSfHH103e, another plasmid important for symbiosis [46,58], these values were 7.4% and 3.6%, respectively. Thus, interestingly, the two plasmids more relevant for symbiosis were likely the replicons showing the highest percentages of induced genes upon non-ionic osmotic stress. The expression of a high number of genes was affected by this condition; thus, it is not possible to comment on all of them. However, it is worth considering the presence of numerous genes involved in ABC transport systems, some induced, others repressed, which indicates that the presence of 400 mM mannitol might alter the transport of solutes across the bacterial envelope. Another group of genes for which transcription was also clearly altered was that coding for chaperonins, especially GroEL which is required for the appropriate folding of hundreds of proteins, approximately 10% to 15% of the cellular proteins [63,64]. Rhizobial species harbour several copies of *groEL* (i.e., there are 5 copies in *S. meliloti* and 3 in *Mesorhizobium huakuii*) [65,66]. Interestingly, all four of the *groEL* copies present in *S. fredii* HH103 that were co-expressed with the coding gene of the co-chaperone GroES were repressed by non-ionic osmotic stress, whereas two additional copies of *groEL* that were not associated with *groES* were induced in that condition. This result suggested that some chaperonins might be more relevant, and thus differentially expressed, for adaptation to non-ionic stress conditions. Clearly, more research is necessary to shed light on this issue.

Recent studies have demonstrated that different osmotic stresses, either ionic (NaCl) or non-ionic (mannitol, dulcitol), activate the expression of *R. tropici* CIAT 899 symbiotic genes, including those involved in NFs (*nod* genes) and those related to nitrogen fixation (*nif* and *fix* genes) [9,12]. In the case of osmotic stress caused by a high concentration of mannitol, we observed strong repression of several *exo* and *kps* genes, which are involved in the synthesis of exopolysaccharide (EPS) and capsular polysaccharide (KPS) respectively, and induction of genes related to the production of cyclin glucans (CG) in *R. tropici* CIAT 899 [12]. On the other hand, in this bacterium, ionic osmotic stress provoked by high concentrations of NaCl also induced the expression of genes related to the production of IAA [9].

In this work. we showed that non-ionic osmotic stress caused by the presence of a high concentration of mannitol had an effect on different physiological traits of *S. fredii* HH103, including the production of different symbiotic molecular signals. Upon this growth condition, this rhizobial strain exhibited enhanced expression of *nod* genes (as scored using the reporter plasmid pMP240 in the β-galactosidase assay and by *q*PCR) and produced significant amounts of biologically active NFs, although showing a lower diversity that those measured upon induction with flavonoids. However, our transcriptomics results did not detect induced expression of the main *nod* genes (*nodABC*) upon non-ionic osmotic stress. Most likely, this may be due to the fact that the RNA-seq experiments were carried out using *S. fredii* HH103 cultures grown in the presence of mannitol for 72 h, whereas both the β-galactosidase assay and qPCR experiments were performed using 16–24-h-old cultures. In fact, preliminary analyses of new RNA-seq experiments following 16 h incubation in the presence of 400 mM mannitol has revealed a slight induction (2–2.5-fold) of *S. fredii* HH103 *nodABC* genes (data not shown). Taking into account all of these data, we hypothesize that the presence of 400 mM mannitol might cause a transient induction of *S. fredii nodABC* genes. In any case, the data provided in this work showed that non-ionic osmotic stress positively influenced the production of NFs by *S. fredii* HH103.

We also observed that non-ionic osmotic stress caused by growth in the presence of a high concentration of mannitol may affect the production of other *S. fredii* HH103 symbiotic signals, since this condition influenced the expression of different genes related to symbiotic T3SS and the production of different surface polysaccharides (EPS, LPS, KPS), as determined by RNA-seq. On the one hand, several genes involved in T3SS were induced. Since the expression of the *ttsI* gene, which codes for the transcriptional activator of this secretion system [35], was very slightly affected (+1.5-fold), one possibility is that the enhanced expression of some T3SS-related genes might be due to the observed repression of *grlR* (−4.9-fold), the product of which is predicted to act as a negative regulator of T3SS. Unfortunately, we could not study whether the secretion of T3Es was enhanced since the presence of 400 mM mannitol negatively affected the extraction of extracellular proteins. Regarding KPS and LPS, the *rkpIJ* genes involved in KPS export [54,67] were strongly induced (+35.2- and 5.6-fold, respectively), whereas several genes putatively involved in LPS production were either induced or repressed. Although the electrophoretic profiles of these polysaccharides were not altered upon growth in the presence of 400 mM mannitol, this methodology was not quantitative and thus the possibility of increased/decreased production of LPS and KPS cannot be discounted. Concerning EPS, we observed the altered expression of several genes, some of them induced, others repressed. Interestingly, *exoR*, of which the *S. meliloti* orthologue has been identified as a repressor of EPS production [68,69], was a repressed gene. This fact might be the cause of the observed enhanced production of EPS by *S. fredii* HH103 when grown in TY medium supplemented with 400 mM mannitol. Furthermore, non-ionic osmotic stress and the presence of flavonoids have opposing effects on EPS production since the production of EPS in *S. fredii* HH103 is repressed by these phenolic compounds in a NodD1- and SyrM-dependent mechanism [70,71]. Finally, it is worth mentioning that the *ndvA* gene, which codes for the protein responsible for CG export from the cytoplasm to the periplasmic space [72,73], was repressed (−5.6-fold) upon growth in the presence of 400 mM mannitol. This fact suggests that *S. fredii* HH103 might accumulate intracellular CGs in response to that stress, but elucidating this possibility would require further investigation.

There are additional *S. fredii* HH103 traits that were influenced by non-ionic stress and might be relevant for symbiosis. Among them is the above-mentioned altered pattern of expression of different *S. fredii* HH103 *groEL* copies. In fact, studies carried out in *S. meliloti* and *M. huakuii* showed that specific copies of *groEL* were essential for symbiosis with their host plants [65,66]. On the other hand, the presence of a high concentration of mannitol clearly affected *S. fredii* HH103 motility, abolishing swimming (most probably because of the increase in viscosity in the culture medium), but promoting surface propagation by a flagellum- and T3SS-dependent mechanism. Interestingly, recent work has shown that the presence of *nod* gene-inducing flavonoids also enhanced *S. fredii* HH103 surface motility, although it did not affect swimming capacity [53]. Flavonoid-induced surface motility also requires the presence of a functional T3SS, but is mediated by both flagellum-dependent and -independent mechanisms. To elucidate whether both surface motility-inducing conditions (presence of *nod*-gene-inducing flavonoids or a high mannitol concentration) share the same flagellum-dependent mechanism would require further investigation. Although rhizobial motility appears not to be essential for symbiosis, it might enhance nodulation efficiency and competitiveness [74,75]. Another interesting trait induced in *S. fredii* HH103 by the presence of a high mannitol concentration was the production of IAA, a phytohormone that has been described as an enhancer of nodule development and symbiotic performance of rhizobia [76]. In fact, many plant growth-promoting rhizobacteria enhance legume–rhizobial symbioses through the production of IAA. Curiously, genistein also enhances the production of IAA in *S. fredii* HH103 through the induction of the same genes (under the transcriptional control of a *nod* box) as mannitol [46,58]. In *R. tropici* CIAT 899, the expression of genes related to the production of IAA is also driven by a *nod* box and induced by flavonoids and ionic osmotic stress, but not in the presence of a high concentration of mannitol [9,12]. In any case, there is the possibility that at least some rhizobia have evolved the ability to produce IAA as a way of improving symbiotic interactions with their host plants. Finally, and very interestingly, the non-ionic stress caused by the presence of 400 mM mannitol provoked a clear increase in the amount and diversity of AHLs produced by *S. fredii* HH103. Most probably, this enhancement was due to the induced expression of the AHL-synthase coding gene *sinI*. In fact, we show in this work that this stress condition acted as a potent elicitor of the production of long chain AHLs in *S. fredii* HH103, which, otherwise, are produced at very low concentrations (non-physiological) when this bacterium is grown under normal culture conditions [33]. Quorum-sensing systems regulate cooperative behaviours in bacteria, and, in the case of rhizobia, many studies show its relevance for symbiosis [77]. In any case, further studies are required to elucidate the relevance of this increased AHL production in the different phenotypes of *S. fredii* HH103 upon growth in the presence of 400 mM mannitol.

In order to distinguish between osmotic and putative effects on growth upon a too high concentration of a readily metabolisable substrate, we plan to extend our studies of non-ionic osmotic stress conditions caused by the presence of non-metabolisable compounds. In any case, similar results have been previously observed with other rhizobia, such as *R. tropici* CIAT 899, by using metabolisable (mannitol) or non-metabolisable (dulcitol) substrates [12].

## 5. Conclusions

In summary, this work demonstrates that non-ionic stress caused by growth in the presence of a high concentration of mannitol provoked notable changes in *S. fredii* HH103, not only in gene expression but also in various bacterial traits, including the production of NFs, other symbiotic signals, as well as other characteristics that might be relevant for symbiosis with legumes. Our work provides new evidence that, as it has been recently reviewed [62], rhizobial responses to stress might play key roles in symbiotic signalling.

## Figures and Tables

**Figure 1 biology-12-00148-f001:**
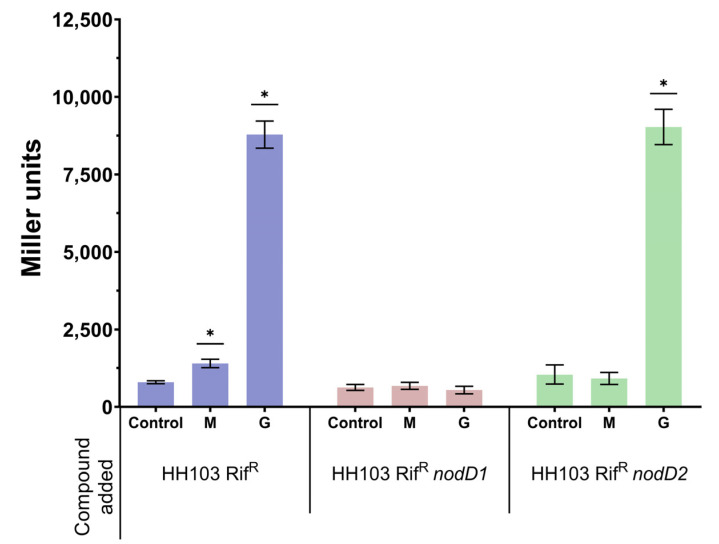
β-galactosidase activity of *S. fredii* HH103 and its *nodD1* and *nodD2* mutant derivatives carrying plasmid pMP240 upon growth in YM (Control) or YM supplemented with 400 mM mannitol (M) or 3.7 µM genistein (G). Asterisks (*) indicate a significant difference from the corresponding control sample using the non-parametric test of Mann–Whitney, α = 5%.

**Figure 2 biology-12-00148-f002:**
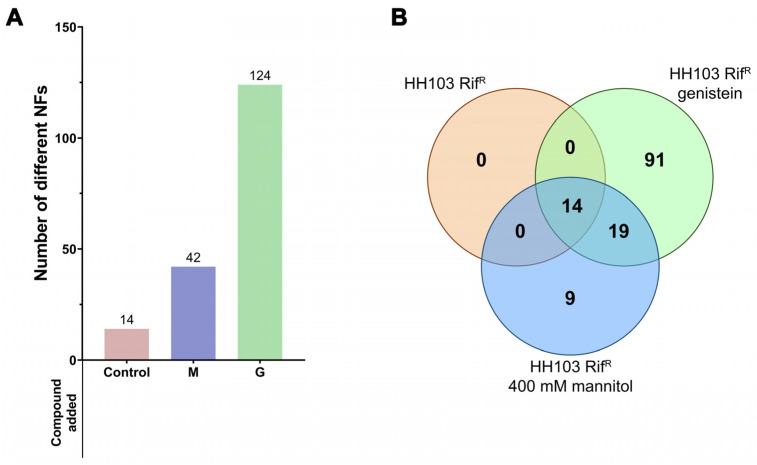
(**A**) Number of NFs produced by *S. fredii* HH103 grown in B^−^ medium in the absence (Control) or presence of mannitol (M) or genistein (G). (**B**) Venn diagram of the sets of Nod factors produced by *S. fredii* HH103 grown in B^−^ medium in the absence (orange circle) or presence of mannitol (blue circle) or genistein (green circle).

**Figure 3 biology-12-00148-f003:**
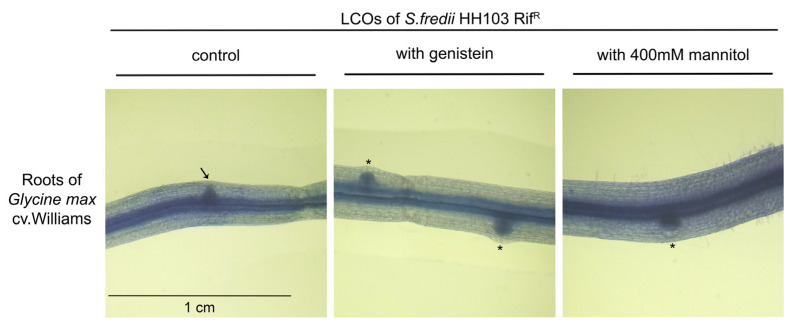
Biological activity assay of the NFs purified from *S. fredii* HH103 cultures grown in B^−^ medium in the absence (control) or presence of 3.7 µM genistein or 400 mM mannitol applied to soybean roots. Nodule primordia (circular shape) are denoted by asterisks, whereas an arrow marks an emerging lateral root (spiky shape).

**Figure 4 biology-12-00148-f004:**
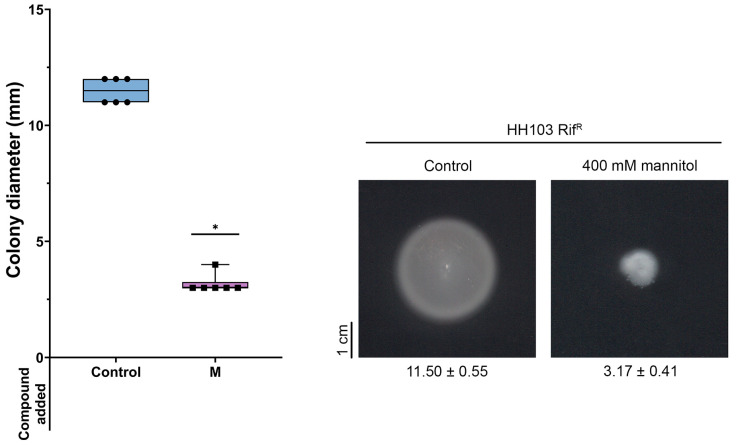
Swimming ability of *S. fredii* HH103 in Bromfield medium in the absence (Control) or presence of 400 mM mannitol (M). All data points are shown in box and whisker plots from at least three biological replicates performed with two technical replicates. Values under images represent the average and standard deviation of migration (given in millimetres and determined as described in the text). Asterisks (*) indicate a significant difference from the corresponding control sample using the non-parametric test of Mann–Whitney, α = 1%.

**Figure 5 biology-12-00148-f005:**
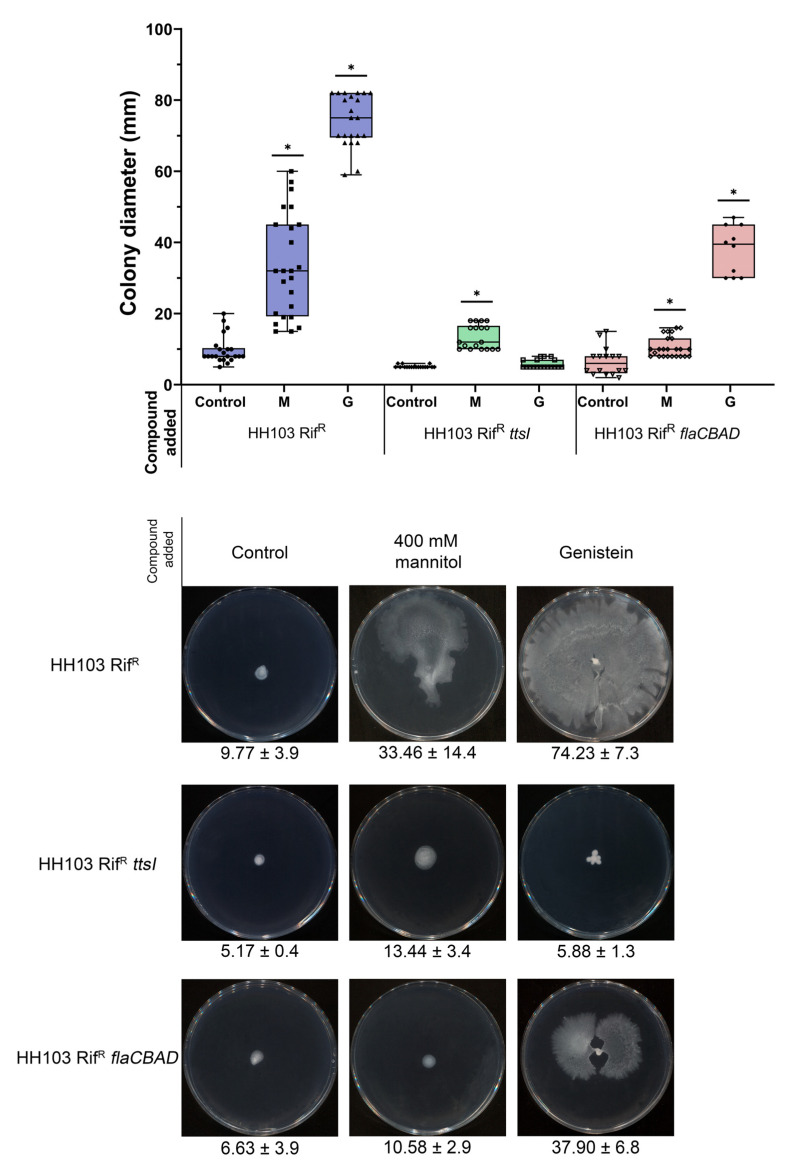
Surface motility of *S. fredii* HH103 and its *ttsI* and *flaCBAD* mutant derivatives in MM in the absence (Control) or presence of 400 mM mannitol (M) or 3.7 µM genistein (G). All data points are shown in box and whisker plots from at least three biological replicates performed with two technical replicates. Values under images represent the average and standard deviation of migration (given in millimetres and determined as described in the text). For each strain, asterisks (*) indicate a significant difference from the corresponding control sample using the non-parametric test of Mann–Whitney, α = 1%.

**Figure 6 biology-12-00148-f006:**
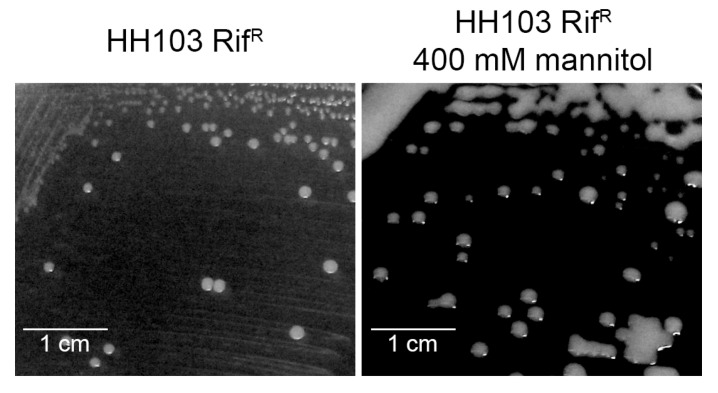
Mucoidy of *S. fredii* HH103 grown in TY medium in the absence or presence of 400 mM mannitol.

**Figure 7 biology-12-00148-f007:**
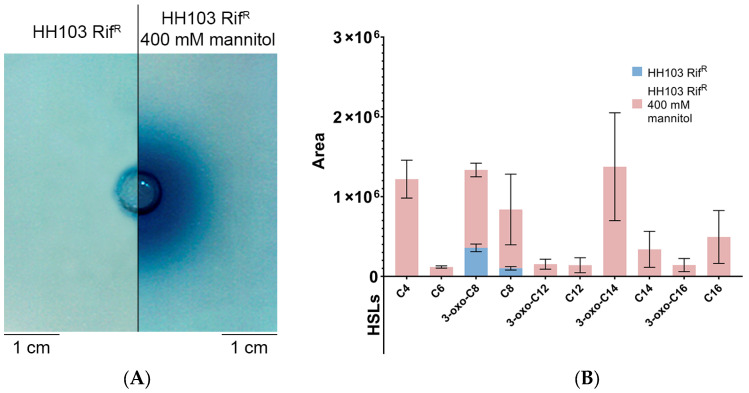
(**A**) Well diffusion assay, using the biosensor *A. tumefaciens* NT1 (pZRL4), for detection of AHLs produced by *S. fredii* HH103 grown in TY medium in the absence (left) or presence (right) of 400 mM mannitol. (**B**) Specific AHLs produced by *S. fredii* HH103 grown in TY medium in the absence (blue) or presence (pink) of 400 mM mannitol, as determined by HPLC-MS/MS.

**Table 1 biology-12-00148-t001:** Selected genes and primer sequences for *q*RT-PCR assays.

Gene Name	Forward	Reverse
*d257*	acagacagctaaattctctgc	gatgttgtcatcctctggata
*ndvA*	cagctacaaccgtatcgaag	gaggatcgtcatcatcgaaat
*nodA*	cgtcatgtatccggtgctgca	cgttggcggcaggttgaga
*nopX*	gcaaatctcctccgtaca	ctcttcaatcgccttcct
*oppD*	gtctcgttccagctctatc	cgccgagaatttcagga
*rkpI*	ggcctctacatgtatttcga	ccggaccgtagaacagcgaga
*SFHH103_1866*	gtcgaagcgttttacattac	gaggaaatccagtccatagc
*SFHH103_3035*	gtggcgactttatggaac	gagttcgatcacctcctc
*SFHH103_3139*	aatgatctgacctggcatt	gctcgatatactcgttgatg
*SFHH103_6032*	cggatctgaaacgcatgt	cgcggaaaataaatgcag
*sinI*	cctgggagatatccggttcga	cactacgacaaggcca
*16S*	gataccctggtagtccac	taaaccacatgctccacc

**Table 2 biology-12-00148-t002:** Distribution of *S. fredii* HH103 DEGs in the presence of 400 mM mannitol among the different replicons in the genome.

Replicon	Number of Genes	Induced	%	Repressed	%
Chromosome	4014	119	2.96	292	7.27
pSfHH103e	1991	148	7.43	72	3.62
pSfHH103d = pSym	667	77	11.54	6	0.90
pSfHH103c	169	4	2.37	9	5.33
pSfHH103b	62	2	3.23	5	8.06
pSfHH103a2	38	0	0	8	21.05
pSfHH103a1	19	0	0	1	5.26
Total	6960	350	5.03	393	5.65

**Table 3 biology-12-00148-t003:** RNA-seq data validation using *q*RT-PCR. The expression of 11 *S. fredii* genes was measured both in the presence or absence of 400 mM mannitol. For *q*RT-PCR, fold-change values were calculated using the ΔΔCt method and normalised to the reference 16S rRNA gene.

		Fold-Change	
Gene	Description	RNA-seq	qPCR
SFHH103_06032	ABC-type sugar transport systems, ATPase components	146.7	33.0
SFHH103_00224, *rkpI*	capsular polysaccharide biosynthesis\export transmembrane protein	35.2	2.5
SFHH103_06488, *oppD*	ATPase components of various ABC-type transport systems	18.8	9.7
psfHH103d_257	Histidinol-phosphate aminotransferase	18.1	7.0
SFHH103_01572, *sinI*	Autoinducer synthase protein	10.8	9.1
psfHH103d_335, *nopX*	T3SS component NopX	6.3	3.0
psfHH103d_126, *nodA*	N-acyltransferase nodulation protein NodA	−1.5	3.9
SFHH103_03433, *ndvA*	Glucan exporter ATP-binding protein	−5.6	−6.3
SFHH103_03035	Conserved hypothetical protein. P pilus assembly/Cpx signaling pathway, periplasmic inhibitor/zinc-resistance associated protein	−29.0	−25.0
SFHH103_01866	RTX toxins and related Ca^2+^-binding proteins	−25.6	−16.7
SFHH103_03139	Oxidoreductase	−58.6	−3.7

## Data Availability

The list of Nod Factors produced by *S. fredii* HH103 under the different conditions studied in this work are available as Appendix A. Fold-changes of *all S. fredii* HH103 genes in the presence of 400 mM mannitol regarding growth in YM medium are available as Appendix A. The list of the different AHLs detected as produced by *S. fredii* HH103 grown in the presence or absence of 400 mM mannitol is available as Appendix A. The RNA-Seq data discussed in this publication can be found in the Sequence Read Archive of NCBI (BioProject database) under BioProject IDs PRJNA911807 and PRJNA313151.

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
