# Peer review of "Non-Ionic Osmotic Stress Induces the Biosynthesis of Nodulation Factors and Affects Other Symbiotic Traits in Sinorhizobium fredii HH103"

_biology, 2023, doi:10.3390/biology12020148_

Round 1

Reviewer 1 Report

This paper describes the effect of growth in 400 mM mannitol on various aspects of the biology of Sinorhizobium fredii strain HH103.  Growth in this condition in several media, which is assumed in this paper to create conditions of non-ionic osmotic stress, led to increases in Nod factor production, ability to form nodule primordial, changes in gene expression, and changes in the ability to make AHLs.  Swimming motility was abolished whereas flagellar dependent surface motility was promoted by 400 nM mannitol. 

Overall, this is quite interesting, though I have a few questions and concerns (see below).  The writing, while certainly understandable, could benefit from thorough proof-reading by a native speaker of English.  Some examples from the very beginning (title and abstracts)......

Title.  Should say Symbiotic, not symbiotical

Lines 21-22. " Recent works have shown that osmotic stress can 21 also promote the formation of symbiotical signals in some rhizobia".  Should read Recent WORK HAS shown ....... SYMBIOTIC signals...

Lines 24-25 " Non-ionic osmotic stress (such as that that can be found 24 in the rhizosphere or inside the legume host  ) WOULD read much better as non-ionic osmotic stress, which can be encountered by the bacterium in the rhizosphere or inside the legume host. 

Line 31.  Also takes, not takes also

Concerns and comments:

1) I was surprised at the choice of a metabolizable substrate to create conditions of osmotic stress.  Would it not have made more sense to use a non-metabolizable organic compound - possibly a sugar or sugar alcohol, but maybe also something like PEG if no suitable sugar or sugar alcohol is available?  Also, was the 55 mM Mannitol already in the media included in the total of 400 mM ?  Since only one condition for induction of osmotic stress was used throughout the paper, it becomes difficult to disentangle what are osmotic effects and what are other effects of growth on a too high concentration of a readily metabolizable substrate. 

2) What is the evidence that the cells are actually stressed osmotically, other than the decrease in growth rate in 3 out of 4 media?  Are any markers of stress such as increase in concentration of typical compensatory solutes (glycine betaine, trehalose ?), changes in Outer membrane proteins and so on available to confirm this ?

3) Why does nodA expression in the RNAseq go down, and only go up slightly in qRTPCR, this would appear to contradict one of the main themes of the paper? (Table 3). 

4) It is stated in lines 593 and 594 that the surface motility induced by 400  mM mannitol is flagellar independent..  This quite clearly wrong, since in figure 5 it is manifestly clear that the flaABCD mutant is not swarming or exhibiting any form of surface motility in 400 nM mannitol, whereas it is in the presence of genistein (bottom panel) 

Author Response

We would like to sincerely thank again the two reviewers for their really useful comments and suggestions and for carrying out their work so quickly.

These are our answers to the different questions posed by the reviewers.

Reviewer 1

This paper describes the effect of growth in 400 mM mannitol on various aspects of the biology of Sinorhizobium fredii strain HH103. Growth in this condition in several media, which is assumed in this paper to create conditions of non-ionic osmotic stress, led to increases in Nod factor production, ability to form nodule primordial, changes in gene expression, and changes in the ability to make AHLs. Swimming motility was abolished whereas flagellar dependent surface motility was promoted by 400 nM mannitol.

Overall, this is quite interesting, though I have a few questions and concerns (see below). The writing, while certainly understandable, could benefit from thorough proof-reading by a native speaker of English. Some examples from the very beginning (title and abstracts)....

Thank you very much for your suggestion. Following your indications, a native speaker of English has carried out a thorough proof-reading of the manuscript.

Title. Should say Symbiotic, not symbiotical.

Corrected

Lines 21-22. "Recent works have shown that osmotic stress can 21 also promote the formation of symbiotical signals in some rhizobia". Should read Recent WORK HAS shown ....... SYMBIOTIC signals...

Corrected.

Lines 24-25 "Non-ionic osmotic stress (such as that that can be found 24 in the rhizosphere or inside the legume host) WOULD read much better as non-ionic osmotic stress, which can be encountered by the bacterium in the rhizosphere or inside the legume host.

Corrected.

Line 31.  Also takes, not takes also

Corrected.

Concerns and comments:

1) I was surprised at the choice of a metabolizable substrate to create conditions of osmotic stress. Would it not have made more sense to use a non-metabolizable organic compound - possibly a sugar or sugar alcohol, but maybe also something like PEG if no suitable sugar or sugar alcohol is available? Also, was the 55 mM Mannitol already in the media included in the total of 400 mM? Since only one condition for induction of osmotic stress was used throughout the paper, it becomes difficult to disentangle what are osmotic effects and what are other effects of growth on a too high concentration of a readily metabolizable substrate.

Thank you very much for your comments. It is true that a non-metabolizable substrate might have been selected for this work in order to differentiate between effects of the stress and effects of the use of high concentrations of a metabolizable substrate. However, we decided to analyze whether the already demonstrated effect of high concentrations of mannitol on the production of symbiotic signals by Rhizobium tropici CIAT899 (del Cerro et al. 2019, ref. number 12 in References) also took place in a different rhizobial strain, S. fredii HH103. We think that a metabolizable substrate might be more probably found in high concentrations either in the rhizosphere or inside the plant that a non-metabolizable one. In the original work carried out by del Cerro and collaborators, although all the analyses were carried out in the presence of 400 mM mannitol, it was also shown that both nod gene expression and NF production were also enhanced in the presence of a non-metabolizable substrate (dulcitol). As we have stated in the new version of the manuscript (please, see lines 940-945 in the tracking changes version), we plan to continue our studies about the effect of non-ionic osmotic stress in S. fredii HH103 and, clearly, this aspect is one to be studied.

Regarding the other question: a final concentration of 400 mM mannitol was employed (the 55 mM Mannitol already present in YM or B- media was included in the total of 400 mM). This was stated in the original version of the manuscript (please, see lines 139-140 in the tracking changes version).

2) What is the evidence that the cells are actually stressed osmotically, other than the decrease in growth rate in 3 out of 4 media? Are any markers of stress such as increase in concentration of typical compensatory solutes (glycine betaine, trehalose?), changes in Outer membrane proteins and so on available to confirm this?

Thank you again for your comments. Considering previous works carried out in rhizobia, we honestly think that growing in the presence of 400 mM mannitol causes osmotic stress in these bacteria. Also, although we have not measured specific markers for osmotic stress such as those detailed in the reviewer’s question, we think that we have detected some changes in gene expression (ABC transport systems, chaperones and other stress protecting proteins, proteins related to the use of trehalose and glycine betaine, etc.) and in bacterial traits (such as partial growth impairment of abolishment of swimming) that might be a good indicative of osmotic stress. These aspects had been commented in the Discussion of the original version of the manuscript.

3) Why does nodA expression in the RNAseq go down, and only go up slightly in qRTPCR, this would appear to contradict one of the main themes of the paper? (Table 3). 

Reviewer is right. This is an intriguing fact. We think that nod genes expression is transiently induced by the presence of 400 mM mannitol. In fact, those experiments carried out after 16-24 hours of incubation in the presence of 400 mM mannitol (β-galactosidase, qPCRs) showed slight induction of nodA, whereas the RNAseq performed using 72-h old cultures did not show significant changes in the expression of this gene. Moreover, preliminary analyses of new RNAseq experiments performed following 16 h of incubation in that condition again revealed slight induction of nodA. In any case, a transient expression of nod genes might account for the enhanced production of NF detected in this condition. This issue had been discussed in the original version of our manuscript (please see lines 854-869 of the tracking changes version).

4) It is stated in lines 593 and 594 that the surface motility induced by 400 mM mannitol is flagellar independent..  This quite clearly wrong, since in figure 5 it is manifestly clear that the flaABCD mutant is not swarming or exhibiting any form of surface motility in 400 nM mannitol, whereas it is in the presence of genistein (bottom panel) 

Reviewer is right. Thank you for advertising us of this misprint. It is true that surface motility induced by 400 mM mannitol is absolutely dependent on the presence of flagella. This mistake has been corrected.

Reviewer 2 Report

Comments to the authors:

The authors analyzed the effects of nonionic osmotic stress on the symbiotic traits of HH103, including the biosynthesis of NFs This is an interesting study, which provide new cues to further uncover the mechanism of establishment of symbiosis between soybean and rhizobia under stress. The MS should be published after revised.

All my questions about this article are as follows:

1.      Please indicate the culture concentration of bacteria, Line 172, for RNA extraction and sequencing.

2.      Please analyze the significance of the data in Figure S1.

3.      Please indicate the significance between Control and 400 mM mannitol (M) in Figure 1.

4.      YM medium supplemented with 1 µM genistein (Line 340), and 400 mM mannitol or 3.7 µ M genistein (Line 350) in the subsequent result description, the concentration of genistein is not the same. Does this mean that the two groups of treatments are different? Please explain

5.      I think these results show that it is not enough to prove that NFs produced when growing in B medium supplemented with 400 mM mannitol has the same biological activity as NFs produced when induced with genistein. Please supplement the expression of some nodule genes at different time points in soybean roots treated with two NFs of different treatments. (Line 392-393)

6.      Please indicate the scale bars in Figure 3.

7. In Figure 6, please add the corresponding scale bars to check whether there is difference in mucoidy size.

Author Response

We would like to sincerely thank again the two reviewers for their really useful comments and suggestions and for carrying out their work so quickly.

These are our answers to the different questions posed by the reviewers.

Reviewer 2

The authors analyzed the effects of nonionic osmotic stress on the symbiotic traits of HH103, including the biosynthesis of NFs This is an interesting study, which provide new cues to further uncover the mechanism of establishment of symbiosis between soybean and rhizobia under stress. The MS should be published after revised.

All my questions about this article are as follows:

  1. Please indicate the culture concentration of bacteria, Line 172, for RNA extraction and sequencing.

Thank you for this observation. The cultures were grown to a final OD600 of 0.8-1.0. This information has been added to the corrected version of the manuscript (please, see line 182 of the tracking changes version of the manuscript). Fortunately, the growing yield obtained in 50 mL Falcon tubes containing 7 ml of YM with a final concentration of 400 mM mannitol was higher than that observed in microtiter plates (Figure S2).

  1. Please analyze the significance of the data in Figure S1.

Thank you for this comment. In the new version of Figure S1 the significance of the data has been analyzed by employing the non-parametric test of Mann-Whitney. A new sentence has been included in the text (please, see lines 296-306 of the tracking changes version of the manuscript) and this information has been added to the figure legend.

  1. Please indicate the significance between Control and 400 mM mannitol (M) in Figure 1.

We apologize for omitting that data. This issue has been corrected in the new version of the manuscript.

  1. YM medium supplemented with 1 µM genistein (Line 340), and 400 mM mannitol or 3.7 µ M genistein (Line 350) in the subsequent result description, the concentration of genistein is not the same. Does this mean that the two groups of treatments are different? Please explain.

Thank you for this observation. We apologize for this mistake. We always use genistein at a final concentration of 1 µg/mL (which is equal to 3.7 µM). We have corrected this mistake thorough the whole manuscript.

  1. I think these results show that it is not enough to prove that NFs produced when growing in B medium supplemented with 400 mM mannitol has the same biological activity as NFs produced when induced with genistein. Please supplement the expression of some nodule genes at different time points in soybean roots treated with two NFs of different treatments. (Line 392-393).

Thank you very much for your very interesting suggestion, which we plan to perform in our next investigations. The original sentence was too pretentious since, at present, we cannot affirm that “the NFs produced upon growth in B- supplemented with a final concentration of 400 mM mannitol are as biologically active as those produced upon induction with genistein”. For this reason, we have changed this sentence. The new version is as follows: “These results clearly indicate that the NFs produced upon growth in B- supplemented with a final concentration of 400 mM mannitol are, as those produced upon induction with genistein, biologically active.” (Please, see lines 414-417 of the tracking changes version of the manuscript).

  1. Please indicate the scale bars in Figure 3.

Done.

  1. In Figure 6, please add the corresponding scale bars to check whether there is difference in mucoidy size.

Done.
